# Magnetic fraction of the atmospheric dust in Kraków – physicochemical characteristics and possible environmental impact

Jan M. Michalik[1], Wanda Wilczyńska-Michalik[2], Łukasz Gondek[1], Waldemar Tokarz[1], Jan Żukrowski[3], Marta Gajewska[3], Marek Michalik[4]

[1] AGH University of Science and Technology, Faculty of Physics and Applied Computer Science, Department of Solid State Physics, A. Mickiewicza Av. 30, 30-059 Krakow, Poland;
[2] Pedagogical University, Institute of Geography, ul. Podchorążych 2, 30-084 Kraków, Poland;
[3] AGH University of Science and Technology, Academic Centre for Materials and Nanotechnology, A. Mickiewicza Av. 30, 30-059 Kraków, Poland;
[4] Jagiellonian University, Institute of Geological Sciences, ul. Gronostajowa 3a, 30-387 Kraków, Poland

*Correspondence to*: Jan M. Michalik (jmichali@agh.edu.pl)

**Abstract.** It is well established that airborne, magnetic nano and microparticles accumulate in human organs (e.g. brain) rising risk of various civilization diseases (e.g. cancer, neurodegenerative diseases). Therefore, precise characterization of that materials, including its origins, is a key factor to prevent their further, uncontrolled emission and circulation. Magnetic fraction of the atmospheric dust was collected in Kraków using a static sampler and analysed using several methods (scanning electron microscopy with energy dispersive spectrometry, transmission electron microscopy with energy dispersive spectrometry, X-ray diffraction, Mössbauer spectroscopy, and vibrating sample magnetometry (VSM) measurements). The magnetic fraction contains magnetite, hematite and $\alpha$-Fe, as well as quartz, feldspar and pyroxene often attached to the magnetic particles. The magnetic particles vary in size from above 20 $\mu$m to nanoparticles below 100 nm, as well as in morphology (irregular or spherical). Their chemical composition is dominated by Fe, often with Mn, Zn, Cr, Cu, Si, Al, S, Ca and other elements. Mössbauer spectroscopy corroborates the composition of the material, giving further indication of smaller than 100 nm particles present in the atmospheric dust. VSM measurements confirm that the strength of the magnetic signal can be treated as a meter of the anthropogenic impact on the suspended particulate matter, once again highlighting the presence of nanoparticles.

## 1 Introduction

Magnetic fraction of atmospheric dust can be considered as a main carrier of metals, especially Fe and transition metals. Usually, the magnetic properties of the total particulate matter samples (e.g. PM10, PM2.5, PM1) are analysed (Górka-Kostrubiec et al., 2020; Morris et al., 1995; Muxworthy et al., 2003; Petrovský et al., 2013; Revuelta et al., 2014; Sagnotti et al., 2006; Spassov et al., 2004; Wang et al., 2017). Baatar et al. (2017) studied the magnetic properties of atmospheric dust removed from the atmosphere during rainfall. The magnetic characteristics of the atmospheric particulate matter is sometimes studied using dust fall samples (e.g. Liu et al. 2019; Magiera et al. 2010). The magnetic properties of particles deposited on

biological surfaces can be useful in terms of the biomagnetic monitoring of atmospheric pollution (Hofman et al., 2017; Maher, 2009; Mejía-Echeverry et al., 2018). Rutkowski et al. (2020) studied the magnetic parameters of dust deposited on spider webs. Magnetic fraction of the atmospheric particulate matter is rarely collected separately using specially constructed samplers (Cheng et al., 2018, 2014; Wirth and Prodi, 1972). Magnetic fraction collected separately, even containing an admixture of non-magnetic particles, allows more precise characterisation of magnetic particles compared with total particulate matter sample (e.g. PM10).

Zhang et al. (2020), determined (after extraction from PM2.5 samples) an annual mean concentration of magnetite nanoparticles (with a relatively broad distribution from 80 to 230 nm) in urban atmosphere in Beijing to be $75.5 \pm 33.2$ ng m$^{-3}$ and daily intake of magnetite was estimated to be $16.6 \pm 7.3$, $29.5 \pm 13$, and $36.4 \pm 16$ ng kg$^{-1}$ of body weight (bw) per day for adults, children, and toddlers respectively. For outdoor professions intake value can reach up to 42 ng kg$^{-1}$ bw day$^{-1}$.

Metal-containing particles are hazardous for human health (e.g., Sorensen, 2005; Zhang et al. 2020). The toxicity of metallic particles is related among others, to the oxidative stress. The effect is significant for transition metal-containing particles because of Haber–Weiss and Fenton-type reactions (Biswas and Wu, 2005; Manke et al., 2013). Morris et al. (1995) proved the correlation between magnetic susceptibility and the mutagenicity of organic extracts from filters containing PM10 particulate matter. Maher et al. (2016) proved that magnetite nanoparticles present in the human brain are of airborne origin. Magnetite pollution nanoparticles may constitute a risk factor for Alzheimer's and Parkinson's diseases (Calderón-Garcidueñas et al., 2019; Gonet and Maher, 2019; Maher, 2019). Calderón-Garcidueñas et al. (2019) identified combustion- and friction-derived iron-rich, strongly magnetic nanoparticles in the hearts of residents of polluted cities. Lu et al. (2020) identified exogenous nanoparticles containing Fe and other transition metals and metalloids in human serum and pleural effusion.

Metals (e.g. Fe, Ti, Mn) in aerosol particles are active in the catalytic oxidation of $SO_2$ in the atmosphere and formation of sulphate aerosol (Alexander et al., 2009; Dupart et al., 2012). Dust acts as a carrier of nutrients (e.g. Fe, P) transported to aquatic environments (Baker et al., 2006; Buck et al., 2010) and consequently can modify processes in the biosphere by providing various elements (e.g. Zn, Cu, Mn) (e.g., Mackey et al., 2015; Mahowald et al., 2018; Paytan et al., 2009). Fe-rich particles are commonly dark coloured and participate in the heating effect in the atmosphere (Moteki et al., 2017). It was evaluated that aggregated magnetite nanoparticles from anthropogenic sources contribute to 4–7 % of the shortwave absorption of black carbon (Ito et al., 2018).

The aim of the present study was to characterize magnetic fraction of aerosols in Kraków. Because collection of the analytical material is very important in such study a simple passive sampler was prepared and used for that purpose.

## 2. Methods

### 2.1 Magnetic fraction collection and experimental protocol

To collect the magnetic fraction of atmospheric dust, a static (passive) sampler composed of a matrix of solid magnets arranged to increase gradients and magnetic field strength was used. It was covered with a 25 μm thick PVC film in order to ease the separation of the collected sample from the sampler itself. Two double collecting surfaces were situated vertically ca. 1.5–1.7 m above the surface of ground covered by grass. The sampling site was situated at the III Campus of Jagiellonian University (Gronostajowa Street, Kraków, Poland) – see figure 1 (please refer to the Supplementary Information I for more detailed information on the sampler and the sampling site). We collected the sample for 9 months (which covers up several seasons). However particles we are aiming to trace are formed either in high temperature industrial processes or may be related to other activities with no seasonal variations (e.g. transport ). Consequently winter/summer differences are not as important as in the case of the investigation of coal or other fuel burning products for household heating (e.g. soot).

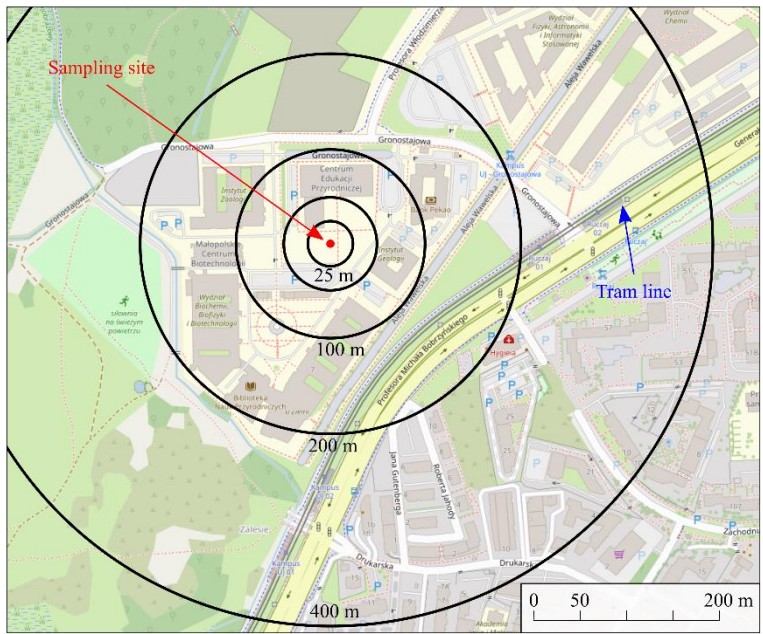

**Fig. 1. Sampling site (50.026916979306854 N, 19.902035577195356 E).**

After the PVC foil was removed from the magnets' matrix it was submerged in isopropyl alcohol and placed in ultrasonic bath in order to detach all the collected material. Then a small fraction of the material was separated for a preparation of specimens for Scanning Electron Microscopy. The rest of the sample was used for consecutive measurements – magnetometry, Mössbauer Spectroscopy and X-ray diffraction (on as collected sample after grounding and on fraction magnetically separated after grinding).

## 2.2 Scanning and transmission electron microscopy with energy dispersive spectrometry

Field emission scanning electron microscope ([FE-SEM], HITACHI S-4700) equipped with X-ray energy dispersive spectrometer ([EDS], NORAN NSS) were used to study the morphology of the components of magnetic fraction and their chemical composition. Samples mounted on adhesive carbon discs were carbon coated. Both secondary electron (SE) and backscattered electron (BSE) modes were used for imaging. Accelerating voltage of 20 kV, 10 μA current and 100 s counting time were applied for the chemical analyses. The standardless method was applied for the quantification of the chemical elements Transmission electron microscopy (TEM) investigations were carried out on a Tecnai TF20 X-TWIN FEG microscope (Thermo Fisher Scientific, working at an accelerating voltage of 200 kV), equipped with an energy dispersive X-ray detector (EDAX). Samples for the TEM observations were prepared by the drop casting of isopropyl alcohol/atmospheric dust dispersion on carbon-coated copper TEM grids. Bright field (BF), scanning transmission electron microscopy (STEM) and high resolution transmission electron microscopy (HRTEM) observations were performed, as well as chemical (EDS) and electron diffraction (selected area electron diffraction [SAED]) analyses.

## 2.3 Mössbauer spectroscopy and magnetisation measurements

[57]Fe Mössbauer measurements were carried out in the transmission mode utilising a constant acceleration spectrometer using [57]Co in a rhodium matrix as a source. Low-temperature measurements were carried out in a cold finger type cryostat, filled with liquid nitrogen. Due to a very low mass of the sample and in order to assure optimal absorbent thickness keeping measurement time reasonable (both meaning the Fe concentration and dimensions) it was mixed with a filler (sucrose was chosen due to ease in its removal before the subsequent measurements). The obtained spectra were fitted using the Gauss–Newton's iterative method of minimising the chi$^2$, with a Lorentzian shape of the spectral lines. Magnetic isothermal loops M(H) at room temperature as well as at liquid nitrogen temperature were measured in a range up to 10 kOe using the Lake Shore Vibrating Sample Magnetometer model 7300 (VSM). The temperature stability was monitored during the measurements. Zero field cooled (ZFC) and field cooled (FC) magnetic curves were measured as well.

## 2.4 X-ray diffraction analysis

X-ray diffraction (XRD) measurements were done by means of Malvern Panalytical Empyrean powder diffractometer using Cu K$_\alpha$ radiation. The powdered samples were placed on single-crystalline silicon zero-background holders. The measurements were performed primarily on the samples without any treatment. Unfortunately, contribution originating from $SiO_2$ surpasses reflections from other phases. Therefore, the samples were grounded and suspended into distilled water in the presence of a magnetic field in order to separate the magnetic fractions, which were of the main interest. This process significantly improved the chances of identifying other phases. The collected diffraction patterns were analysed in terms of the Rietveld method using the FullProf Suite Package (Rodríguez-Carvajal, 1993). FullProf Suite was solely used for Rietveld refinement, which resulted

in quantification of observed crystalline phases. The phase identification was a multistep process, which involved: analyses of chemical composition followed by trial refinements of the most apparent, common phases; usage of automatic phase recognition software (eg. Panalytical HighScore 3.0) for search of secondary phases and confirming the main phases. The

115    resolution of the instrument calibrated using NIST 660 standard is 0.065° of 2θ, which is much below widths of lines in samples. The data were collected with angular step of 0.016° of 2θ. The calibration measurements using NIST 660 revealed also instrumental broadening of the collected lines, which is essential to further analyses of reflections profiles.

## 3 Results and discussion

120    **3.1 General description and mineral composition of the magnetic fraction**

After nine months of exposition, the surfaces of the sampler were covered with a thin and uneven layer of dust (Fig. 2).

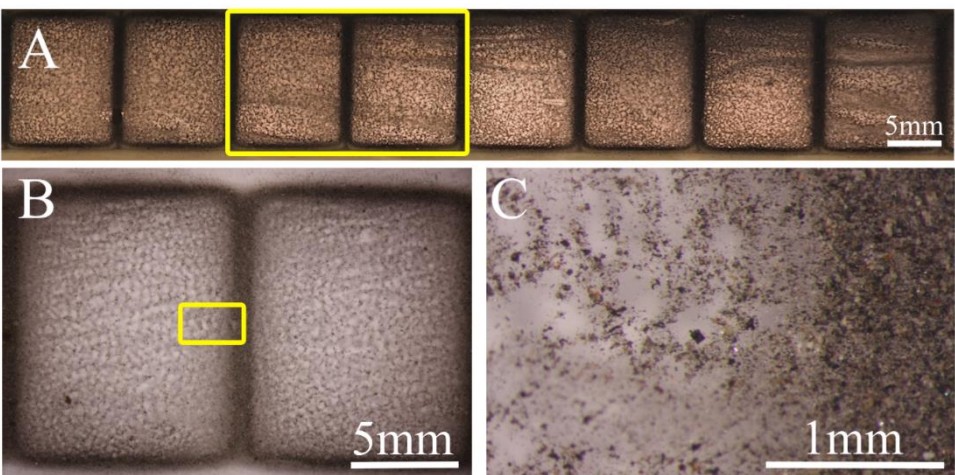

**Fig. 2. Magnetic fraction on the passive sampler after 9 months of deposition; A, B & C. Sample on the surface of rectangular magnet**
125    **covered with PVC foil with dust grains of various colour, size and shape on the surface of the sampler. C&B are zoomed areas marked by yellow rectangular areas in A&B respectively.**

The magnetic fraction collected on the sampler surfaces is composed of grainy material of different size and colour. The dominant part of the grains is dark grey, although colourless and transparent, brown, reddish or lustrous are also present (Fig.

130    2C). The size of the particles observed in electron microscope varies from more than 30 μm to below 100 nm. The number of the largest particles in the sample is low, but their share is significant considering the volume or mass of these particles. Because of the relatively high content of particles larger than 10 μm, the sample differs from typical PM10 samples.

The results of the XRD studies (Fig. 3) suggest that the separated fraction is dominated by magnetite (27.9 wt%), hematite (14.8 wt%) and α-Fe (1.5 wt%) in terms of the magnetic phases. It also exhibits quartz (41.2 wt%), feldspar (10.0 wt%) and pyroxene (4.6 wt%). Quartz grains observed using SEM are angular, different in size (with the largest up to 50 μm). Angular grains with chemical composition similar to K-feldspar and usually platy Ca sulphate crystals (possibly gypsum) were also noted using the SEM-EDS method. Relatively common are grains containing Fe, Si, Al, Ca and other elements (possibly aluminosilicates).

Precise analysis of the profile of magnetite reflexions in the XRD pattern suggest the distribution of various elements at the Fe-sites (e.g. Cr, Mn, Co, Zn) as typical of naturally abundant ferrites (Fig. 3) as evidenced by a strain in the profile of this phase. The strain can be extracted from instrumental broadening (thanks to calibration measurements), therefore Rietveld refinement revealed strains for all phases. The residual strains are related to static defects of the structure eg. atomic disorder on Wyckoff positions originating from substitution of different types of atoms into specific sites. The refined strain was found to be about 0.75%, which is significantly higher than typical values for pure $Fe_3O_4$ specimens which is smaller than 0.2% (The pure $Fe_3O_4$ specimen was also measured as a reference). The resulting broadening of the main reflections of the $Fe_3O_4$ (220 at 30.1° of 2θ and 311 at 35.5° of 2θ) can be noticed in the inserts to the Fig. 3.

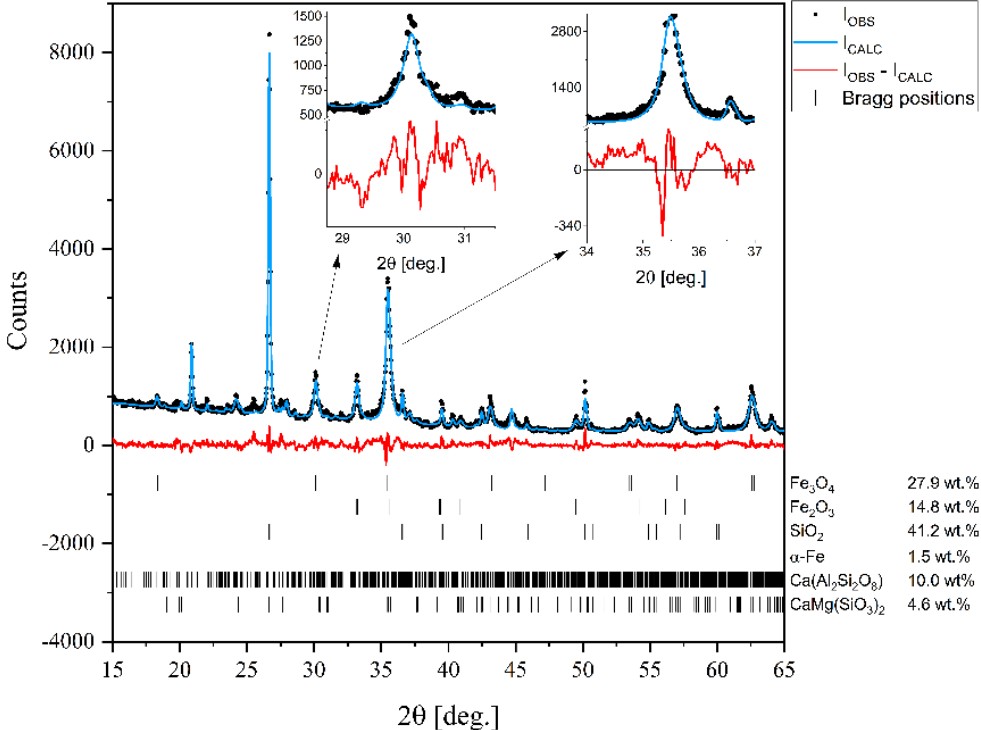

**Fig. 3. X-ray diffraction pattern measured at room temperature. The black circles stand for measured data, the blue line is calculated spectra obtained assuming material composition as indicated, and the red line denotes the difference between the observed and calculated intensities. Calculated positions of the diffraction peaks for each phase are indicated as bars.**

## 3.2 Form of occurrence of Fe-containing particles

Fe-containing particles analysed using the SEM-EDS method differ in size and morphology. Both irregular and spherical particles are present (Fig. 4A). The spherical particles are considered to be of anthropogenic origin and related to high temperature processes. The occurrence of natural spherical particles (cosmic, volcanic, lightning induced; e.g. Genareau et al. 2015; Genge et al., 2017) is very limited in comparison with the anthropogenic ones, but it is not possible to exclude the occurrence of spherical micrometeorites in the dust collected in urban environments (Genge et al., 2017). In the atmospheric particulate matter studied by Ebert et al. (2000) and Choël et al. (2007), most of the Fe-rich particles were spherical, that is, anthropogenic in origin. In the magnetic fraction collected in Kraków, irregular angular particles significantly prevail over spherical ones (especially in the more coarse-grained fractions), taking into account the number of particles. In the case of irregular particles, the distinction between natural and anthropogenic is difficult.

Fe-containing particles occur as discrete forms of diverse size and morphology (Fig. 4A, B). Numerous Fe-containing particles occur as grains attached to the surface of larger grains (e.g. quartz, feldspar, various aluminosilicates, gypsum, spherical particles of various composition and pollen grains) (Fig. 4B–D). The number of Fe-containing particles attached to above mentioned larger grains is strongly variable. This form of occurrence of the atmospheric dust can be considered as an example of heterogenous clustering (Pietrodangelo et al., 2014).

Fe-containing particles also occur as a component of aggregates of various sizes and morphologies (Fig. 4E and F). The size and composition of the particles in aggregates are differentiated. Aggregates of larger particles are heterogenous (heterogenous clustering; (Pietrodangelo et al., 2014)). Homogenous aggregates (homogenous clustering; (Pietrodangelo et al., 2014)) are usually composed of small (below 200 nm) Fe-rich spheres (Fig. 4G and H). Magnetic Fe-containing particles attached to larger grains or present in aggregates causes the accumulation of quartz, feldspars and other non-magnetic components in the magnetic fraction, as evidenced by XRD.

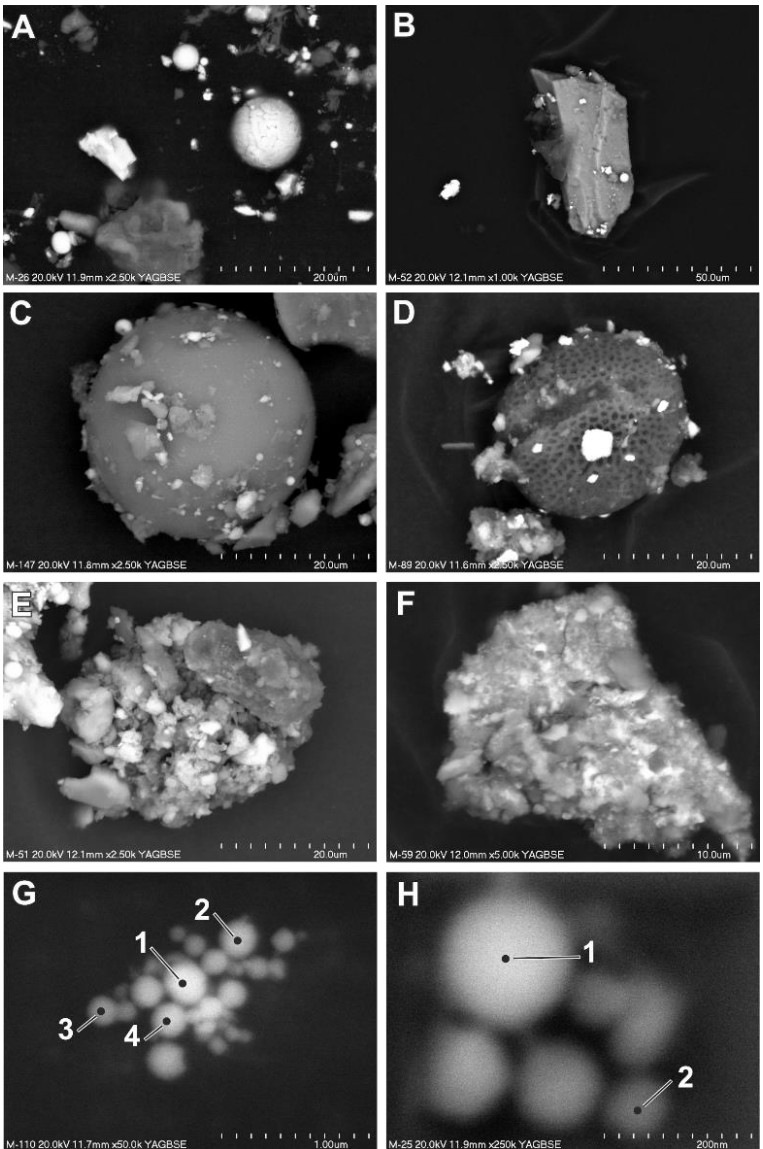

**Fig. 4. Forms of occurrence of magnetic particles (SEM; backscattered electron images). A. Discrete particles of different size and shape; B. Ca-rich aluminosilicate grain with attached magnetic particles; C. Aluminosilicate sphere with attached magnetic particles; D. Pollen grain with attached magnetic particles; E. Aggregate of angular aluminosilicate grains of various chemical composition and Fe-rich particles; F. Aggregate of fine particles often with diffused boundaries; bright particles are Fe-rich; G. Aggregate composed of spherical Fe-rich particles below 200 nm in size, ; H. Aggregate composed of spherical Fe-rich particles below 200 nm in size; (see table 1 for the results of the chemical composition measurements of the indicated particles).**

### 3.3 Size, morphology and chemical composition of Fe-rich particles

The Fe-containing particles vary in size from above 20 μm (rarely) (Fig. 5A and B) to nanoparticles below 100 nm (Fig. 4G, 4H). The particles are irregular in shape, sometimes angular, or spherical (Fig.4A, 5A - D). The content of the spherical particles in fractions above 10, 10–5, 5–2.5 and 2.5–1 μm is variable and significantly lower than the irregular ones. In the

fraction below 1 μm, the number of spherical particles (53 %) slightly dominates the irregular. The results of the chemical analyses of 278 Fe-containing particles (see Supplementary Information II for a complete list of results) indicate that the Fe content varies from 2.32 wt% to 98.18 wt%. Particles with lower content of Fe are usually enriched in Si and Al. For some of them, the chemical composition corresponds to pyroxenes determined using XRD analysis. Trace elements (content below 0.1wt%) noted in single particles are not discussed because of insufficient statistical significance of these results.

Numerous irregular Fe-based particles contain Cr (up to 28 wt%), Zn (up to 19 wt%), Mn (up to 11 wt%) and Cu (up to 7.5 wt%) (Fig. 5H). Ni (up to 8 wt%) occurs rarely, and exclusively in particles containing Cr. W and V occur rarely in the irregular Fe-rich particles, usually in low amounts (W up to 5.45 wt%, V up to 0.76 wt%). Sb and Sn were noted only in a few particles (up to ca. 1.5 wt%). Determination of the origin of the Fe-containing particles is difficult. According to Bogacki et al. (2018) re-entrained road dust contribute to 25 % in winter and 50 % in summer of the PM10 in the air in selected streets in the centre

of Krakow. It indicates that long-lasting, multi-stage evolution and mixing of atmospheric particulate is possible. The form of occurrence (irregular) can be related both to natural sources (e.g. soils and the disintegration of rocks) or anthropogenic ones (e.g. metallurgical industry, fuel combustion, other industrial sectors, and traffic-related sources (exhaust and non-exhaust emissions).

According to Li et al. (2021) iron and steel production is a main source of magnetic particles emission, while emission from
power plants is in the second place. It can be assumed that for studied irregular particles natural origin is less probable, especially for the larger particles where the range of transportation in the atmosphere is limited. However, taking into account their abundance in the atmospheric particulate matter samples and the scarcity of their possible natural sources in the study area, it is possible to assume that the dominant part is of anthropogenic origin. Chemical composition can be considered as an important indication of the origin of Fe-rich airborne particles (cf. Wilczyńska-Michalik et al., 2020a). It is often assumed that
anthropogenic Fe-rich particles are mostly spherical (e.g., Choël et al., 2007). It is the common form originating from high temperature processes, but in iron metallurgy dust of different shape, size and chemical composition can be emitted (Jarzębski and Kapała, 1975; Wilczyńska-Michalik 1981; Jabłońska et al. 2021). In 1979 Steelworks in Krakow alone emitted dust containing 18 000 tons of Fe (Cole 1991). In 1985 emission of Fe in dust in Krakow region was estimated to be 14 000 t (Helios Rybicka 1996). Recently the emission of Fe-rich dust iron metallurgy is significantly lower but still this emission is
present. It is also likely that irregular Fe-containing particles are derived from fragmented metallurgical slags that show variety of chemical and mineral composition, but often contain Fe-rich components (Neuhold et al., 2019; Potysz and Kierczak, 2019), and also Cr and Mn (Horckmans et al., 2019). Metallurgical slags are often used as a substitute of natural aggregate (Horckmans et al., 2019), which could be a reason for the broad dispersion of slag-derived dust in the atmosphere. Irregular Fe-rich particles are also related to rail transportation (Moreno et al., 2015). Most of the Fe-rich particles described in the literature from this
source are composed of hematite. The sampling site was situated ca. 250 m from two tram stops, which indicates that this source could also contribute to the collected magnetic fraction. Fe-rich particles occur commonly in road dust and their origin can be related to non-exhaust traffic emissions such as brake-wear emission (Grigoratos and Martini, 2015).

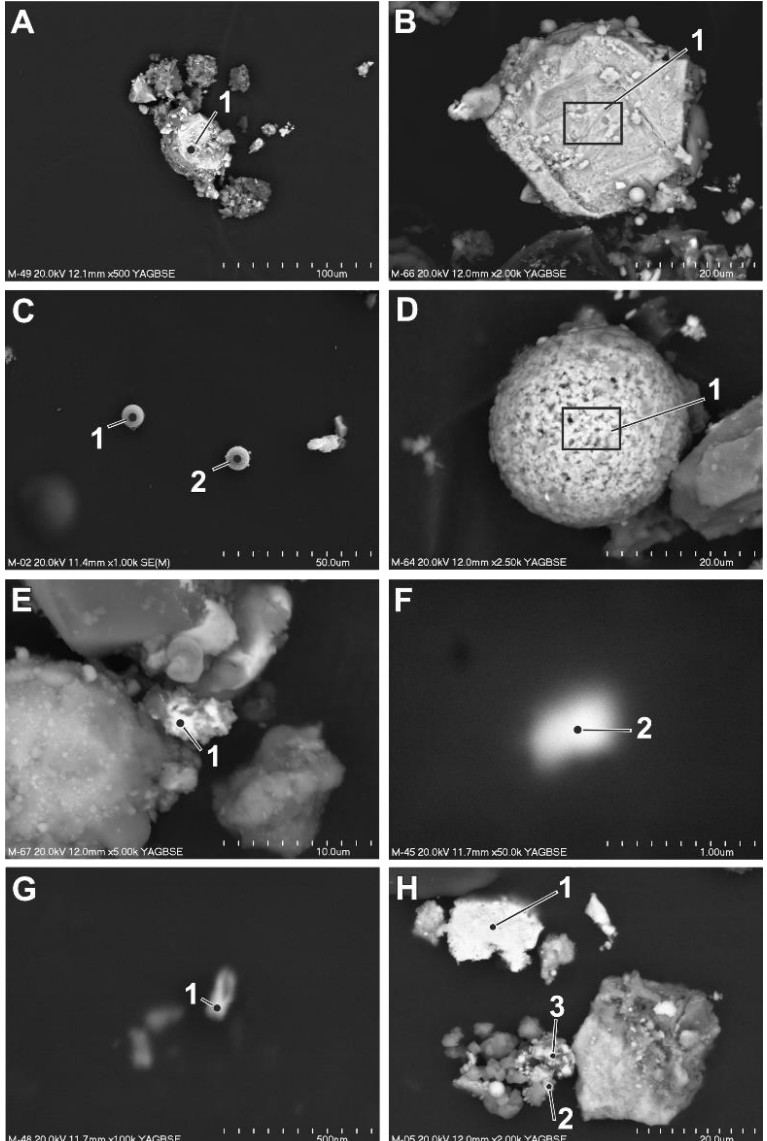

Fig. 5. Size and morphology of Fe-rich particles and particles dominated by other than Fe metals or with high content other elements (SEM; backscattered electron images except C – secondary electrons). A. Irregular particles bigger than 20 μm; B. Irregular particles bigger than 20 μm; C. Aluminosilicate sphere (left) and Fe-dominated sphere (right); D. Spherical particle rich in Ca and Fe; E. Particle rich in Pb; F. Particle rich in Zr; G. Particle rich in Au; H. Particles with high content of other than Fe metals; (see table 1 for the results of the chemical composition measurements of the indicated particles)..

Several types of spherical Fe-rich particles were noted in the magnetic fraction (Fig. 5C). Aluminosilicate spheres containing usually 5–20 wt% of Fe occur rarely (Fig. 4C). Production of energy in coal-fired power plants can be considered as the main source of particulate matter of this type (Wilczyńska-Michalik et al., 2020b). Spherical forms with Fe content within the range of 35–60 wt% and Si in the 15–35 wt% range are not numerous. Only one Fe-rich spherical particle with relatively high content of Ca (possibly calcium ferrite) was noticed (Fig. 4D). Oxides of $Fe_xO_y$ type strongly dominate the group of spherical particles.

Most of these particles can be attributed to the industrial metallurgical processes (e.g., Choël et al., 2007). Spherical particles containing Cr are rare. Zn is also rare in the Fe oxide spherical particles of diameter larger than 1 μm.

Most of the spherical particles below 1 μm contain Zn (up to 27.8 wt%) (Fig. 4G, 4H). This can be related to the deposition of Zn on the surfaces of small particles during the cooling of volatile substances from metallurgical processes (Ebert et al., 2000). Mn is also a common element in Fe-rich spherical particles below 1 μm in diameter, with the highest content around 10 wt%.

Cr occurs relatively rarely in this group of spherical particles. Spherical particles below 200 nm are relatively common in the group of particles below 1 μm. Spherical Fe-rich particles of diameter below 1 μm are formed commonly in metallurgical processes (Jenkins, 2003; Jenkins and Eagar, 2005). Spherical Fe nanoparticles derived from the in-cylinder melting of metallic engine parts contain Mn and Cr (Liati et al., 2015). Rail transport was recognised as a possible source of spherical nanoparticle aggregates composed of magnetite (Moreno et al., 2015). Spherical forms of 500 nm usually occur in clusters.

In the magnetic fraction particles with high content of other metals or dominated by other than Fe metals occur rarely (Fig. 5E, F, G, H). A few particles rich in Pb were noted (up to 70 wt%) (Fig. 5E). Three irregular particles below 1 μm in size devoid of Fe and composed predominately of Zr and O (two particles) (Fig. 5F) or Au (Fig. 5G) were analysed.

S is commonly noted in the studied Fe-rich particles and is often accompanied by Na (Fig. 4H), Cl, K, Ca, Mg and Ba. It was noted by Ito et al. (2018) that Fe in aged fly ashes is coated by Fe sulphates. According to Li et al. (2016), atmospheric metal
particles are internally mixed with secondary sulphates or other components.

A few particles are relatively rich in S and Fe, but without any measured O content that can indicate the presence of sulphide component.

TEM investigations were aimed at the characterisation of the smallest fraction of the analysed atmospheric dust samples. Figure 6 presents examples of bright field TEM images of the studied material. The observed particles vary widely in size and
shape. The smallest observed particulates are well below 10 nm in size, with the large ones exceeding a few hundred nanometres. Irregular morphology is predominant regardless of the particle size (Fig. 6A, E, F, G). Nevertheless, spherical particles of 50–200 nm diameter were also observed (Fig. 6A, B, C, D). Most of the particles analysed using EDS are rich in Fe and O, while some of them also contain Si, Zn, Mn, Al or Mg.

Selected area electron diffraction (SAED) collected from an agglomerate of the smallest fraction particles (Fig. 6E, F) allowed
the determination of a presence of magnetite (JCPDS card no. 00-001-1111), which is consistent with the performed XRD analysis. Domains with the magnetite ordering reach the size of 10 nm (Fig. 6G, H).

HRTEM imaging allowed for proving the existence of the ferrous nanoparticles below 10 nm in diameter on one hand and bigger - up to 200 nm on the other. Moreover we clearly show crystallinity of those smallest particles proving their chemical composition which is an important information to assess their impact to the human health.


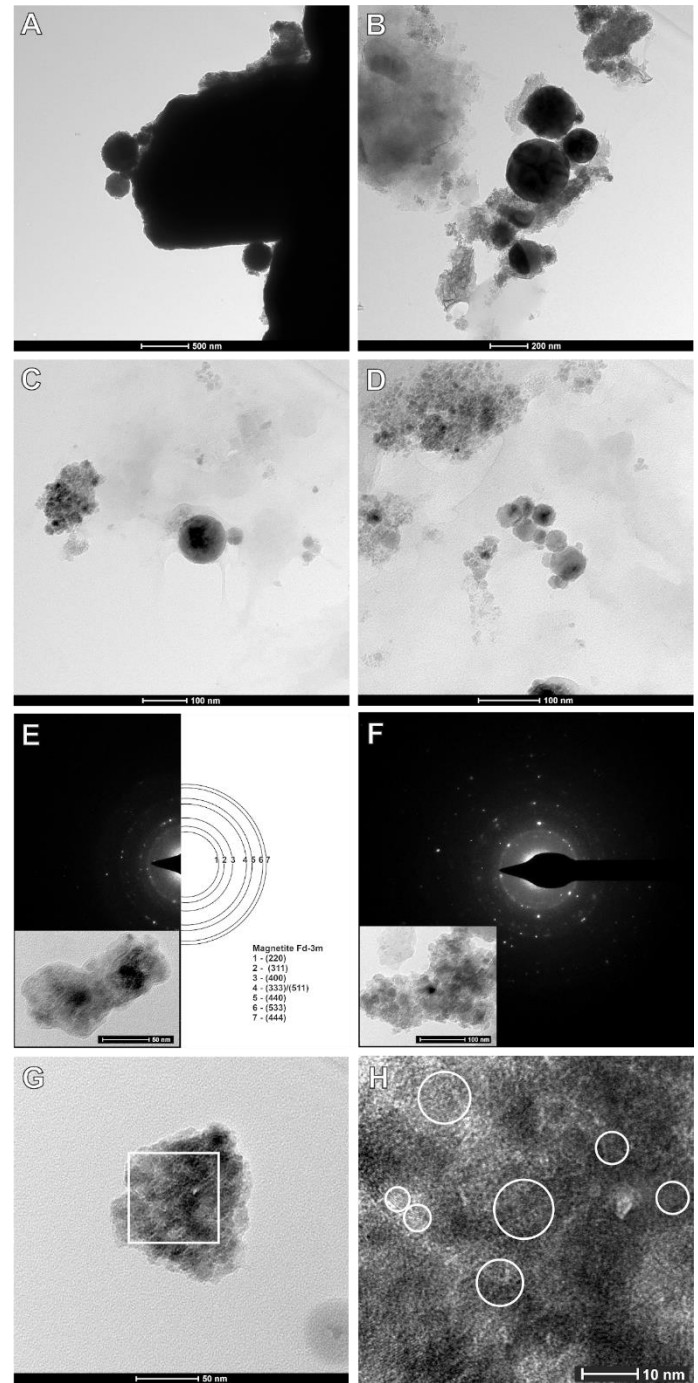

**Fig. 6. Fe- rich particles (TEM studies). A. Large spherical and irregular particles rich in Fe; B, C and D. Spherical particles of various size rich in Fe; E and F. SAED analyses of aggregates of irregular Fe rich particles (particles shown in insets); G. Aggregate of Fe rich particles. White square – area of HRTEM analysis. H. HRTEM analysis; white circles - domains with visible magnetite ordering.**

Table 1. Results of chemical analyses for spots or areas marked at Figures 4 and 5 (empty fields – elements not determined).

| Fig. | Spot or Area | Element (wt. %) | | | | | | | | | | | | | | | | | | | |
|---|---|---|---|---|---|---|---|---|---|---|---|---|---|---|---|---|---|---|---|---|---|
| | | Fe | Mn | Cr | Ni | Zn | W | Pb | Au | Zr | Ti | Si | Al. | P | O | S | Ca | Mg | K | Na | Cl |
| 4G | 1 | 60.04 | | | | 14.66 | | | | | | 3.23 | 1.58 | | 16.27 | 3.40 | 0.82 | | | | |
| | 2 | 62.96 | | | | 3.59 | | | | | | 3.13 | 0.90 | | 17.95 | 3.78 | 1.17 | | | 6.52 | |
| | 3 | 59.38 | 1.35 | | | 4.41 | | | | | | 3.75 | 1.47 | | 18.24 | 4.93 | 1.40 | | | 5.07 | |
| | 4 | 60.19 | | | | 11.67 | | | | | | 3.85 | 1.74 | | 17.53 | 4.04 | 0.98 | | | | |
| 4H | 1 | 62.06 | | | | 3.95 | | | | | | 2.83 | | | 17.29 | 6.29 | | | | 7.58 | |
| | 2 | 46.52 | | | | | | | | | | 5.84 | | | 19.89 | 13.06 | | | | 14.69 | |
| 5A | 1 | 98.18 | | | | | | | | | | 0.50 | | | 0.82 | | 0.34 | | | 0.16 | |
| 5B | 1 | 91.33 | 0.46 | | | | | | | | | 0.56 | | | 7.12 | | 0.53 | | | | |
| 5C | 1 | 4.85 | | | | | | | | | 1.07 | 35.37 | 18.60 | | 31.23 | | 0.54 | 1.79 | 5.00 | 1.55 | |
| | 2 | 87.55 | | | | | | | | | | 0.26 | | 0.13 | 12.06 | | | | | | |
| 5D | 1 | 40.01 | 2.89 | | | 1.47 | | | | | 0.70 | 2.32 | 1.95 | 0.13 | 14.37 | 0.15 | 35.57 | 0.31 | | | 0.13 |
| 5E | 1 | 3.74 | | | | | 2.31 | 70.25 | | | | | | | 11.89 | 10.32 | 1.49 | | | | |
| 5F | 1 | | | | | | | | | 73.08 | | | | | 21.41 | 3.64 | | | | 1.87 | |
| 5G | 1 | | | | | | | | 85.71 | | | 4.05 | | | 0.10 | | | | | 10.24 | |
| 5H | 1 | 68.34 | 5.74 | 12.44 | 6.33 | | | | | | | 3.34 | 0.22 | | 2.72 | 0.49 | 0.38 | | | | |
| | 2 | 55.64 | 4.15 | 0.87 | | 13.61 | | | | | | 4.06 | 2.32 | | 12.93 | 2.22 | 2.64 | 1.56 | | | |
| | 3 | 44.85 | 8.56 | 1.26 | 1.08 | 19.78 | | | | | 0.16 | 2.78 | 0.97 | | 12.70 | 1.19 | 3.37 | 2.91 | | | 0.39 |

## 3.4 Mössbauer spectroscopy

Experimental data collected at room temperature and at 80K were refined using doublet and sextet components (Fig. 6) based on the samples' composition obtained on the basis of XRD and chemical analysis, as well as the available literature data. For the analysis of the results isomer shift (IS), quadrupole splitting (QS, defined as half of the distance between the doublet peaks in our fits) and hyperfine magnetic field (B) were taken into account, together with the relative intensity (rel. int.) of each component. One has to keep in mind that in the case of the material under investigation, we cannot discard the possibility of the particles being of non-stoichiometric composition, having a large number of defects and finally their surfaces being affected by atmospheric conditions.

At room temperature, satisfactory results are obtained using two doublet contributions and three magnetically split sextets. The former ones are attributed to $(Ca,Mg,Fe)(SiO_3)_2$ (20.3 %) and to a collapsed spectra of $FeO_x$ (32.2 %). Sextet due to a hyperfine magnetic field can be ascribed to a presence of $\alpha\text{-}Fe_2O_3$ (18.9 %), large $Fe_3O_4$ (32.4 %) grains and bigger particles, as well as FeX alloy (5.3 %). Low temperature measurements further support our predictions made on the basis of the room temperature measurements showing again two quadrupole split contributions: $(Ca,Mg,Fe)(SiO_3)_2$ (20.2 %) and superparamagnetic iron/iron oxide particles. Three magnetically split components are again related to FeX alloy 7.4 %), $\alpha\text{-}Fe_2O_3$ (10.1 %) and $Fe_3O_4$ grains (35.6 %). As can be seen in Table 1, the contribution from $Fe_3O_4$ changes from two to four sextets. This is due to the fact that in $Fe_3O_4$, the iron cations occupy two inequivalent crystallographic positions: tetrahedral (T) and octahedral (O) sites (Tong et al., 2001). Therefore, the spectrum of $Fe_3O_4$ at room temperature (well above the Verwey transition – 120K) is composed of only two sextets corresponding to $Fe^{3+}$ cations on a T site, and to $Fe^{2+}/Fe^{3+}$ cations on an O site. Below 120K, the relaxation time is extended and consequently, the O sites occupied by $Fe^{2+}$ and $Fe^{3+}$ can be differentiated (Řezníček et al., 2017).

Interestingly the contribution of the superparamagnetic particles is significantly diminished (falling from 24 % at room temperature to less than 5% at 80K). At the expense of the doublet component, a new magnetically split contribution arises related to tiny $Fe_3O_4$-like particles giving about 22 % of the spectra. Such behaviour of the Mössbauer spectrum typically belongs to nanosized ferromagnetic or antiferromagnetic particles. Such particles (of the sizes below a single domain boundary) will present reduced hyperfine fields owing to spin relaxation effects (Yamada and Nishida, 2019). Our Mössbauer spectroscopy results indicate that iron bearing particles in the aerosols are very small, showing a superparamagnetic behaviour at room temperature (RT) and becoming ferromagnetic while lowering the measurement temperature. On one hand, a possibility of superparamagnetic particles being present in the suspended particulate matter was discarded by some authors on the basis of the magnetisation measurements (Magiera et al., 2021). On the other, the possibility of the long-range transportation of tiny (a few nanometres to a few tenths of a nanometre in diameter) particles far from the industrial sources cannot be eliminated. Moreover, such particles will not only travel over large distance, but also stay suspended for a long time. Our results are also in agreement with those presented by other groups studying aerosols and suspended particulate matter (Fu et al., 2012; Muxworthy et al., 2003).

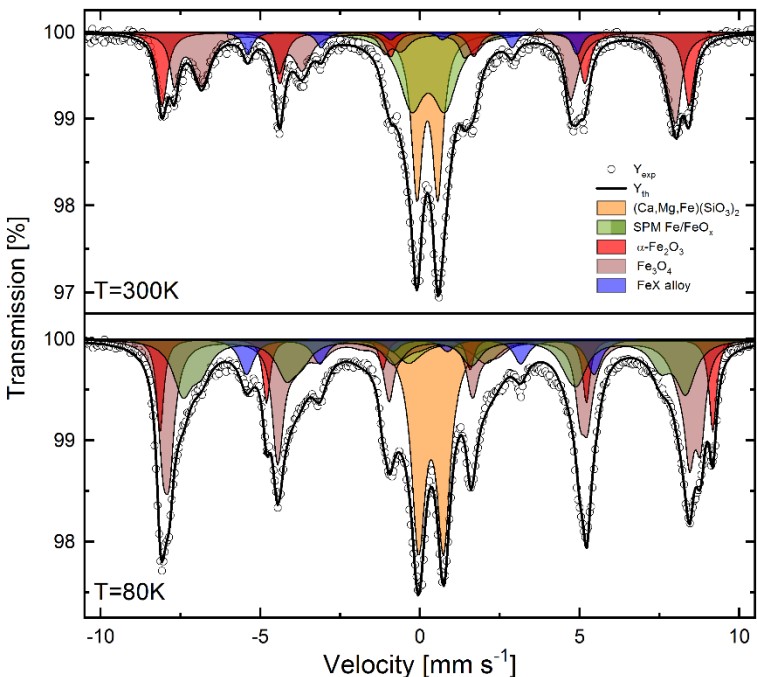

**Fig. 6. Mössbauer spectra measured at room temperature (top) and at 80K (bottom). For detailed explanation and the hyperfine interaction parameter of the components used for fitting the spectra, see the text and Table 2.**

A presence of some amount of aluminosilicates cannot be discarded based on the existence of a small component that is not magnetically split down to 80K; however, the IS values do not fully support this statement, at least for ideal chemical composition and crystalline structure.

We are also aware of the possibility of $Fe^{3+}/Fe^{2+}$ containing glasses to be used for Mössbauer spectra interpretation in the case of coal combustion products (Huffman et al., 1981; Murad and Cashion, 2004). These are commonly found in ashes from high temperature processes (their content getting larger with combustion temperature rise) and are identified on the basis of the QS/IS ratio as they show a sextet collapsed into a doublet with a very broad lines. Taking into account the way we collected our samples, we would rather discard ferric or ferrous glasses from being present in the material under investigation.





**Table 2. Hyperfine interactions parameters (isomer shift, quadrupole splitting, hyperfine magnetic field) and relative component contribution for samples measured at room temperature (300K) and low temperature (80K) (IS with respect to metallic Iron at 300K).**

| Component | Temperature [K] | Rel. Contr. [%] | IS [mm/s] | B [kGs] | QS [mm/s] |
|---|---|---|---|---|---|
| α-Fe$_2$O$_3$ | 300 | 18.9 | 0.389 | 511.4 | -0.105 |
| | 80 | 10.1 | 0.4678 | 536,4 | 0.156 |
| Fe$_3$O$_4$ | 300 | 32.4 | 0.264 | 487.5 | 0.018 |
| | | | 0.645 | 455.4 | -0.026 |
| | 80 | 35.6 | 0.479 | 521.5 | 0 |
| | | | 0.4082 | 504.8 | 0 |
| | | | 0.5446 | 496.1 | -0.325 |
| | | | 0.978 | 393.0 | 1.219 |
| FeX alloy | 300 | 5.3 | -0.065 | 319.6 | -0.067 |
| | 80 | 7.4 | 0.1179 | 337.1 | 0 |
| (Ca,Mg,Fe)(SiO$_3$)$_2$ | 300 | 20.3 | 0.341 | | 0.327 |
| | 80 | 20.2 | 0.4549 | | 0.393 |
| SPM Fe/FeO$_x$ | 300 | 20.3 | 0.369 | | 0.522 |
| | | 2.9 | 1. 258 | | 0.35 |
| | 80 | 22 | 0.518 | 488.1 | 0.037 |
| | | | 0.5096 | 446.0 | -0.060 |
| | | 4.7 | 0.9662 | | 1.212 |

## 3.5 Magnetisation (VSM) measurements

Magnetisation (VSM) measurements of the magnetic fraction were performed between 295 K and 77 K. The sample shows typical ferromagnetic behaviour (Fig. 7). The material is almost fully saturated both at high and low temperatures. At 77 K, a wider hysteresis loop is observed as compared to the RT measurements. The widening of the hysteresis loop could be associated to blocking of superparamagnetic particles at low temperatures. Typically, a well-defined maximum in the ZFC curve is associated to blocking temperature of superparamagnetic nanoparticles of well-defined size. However, for broad distribution of nanoparticles' sizes such maximum is usually not observed. This is apparently our case, as the ZFC curve shows no maximum but constant raise up to the room temperature.

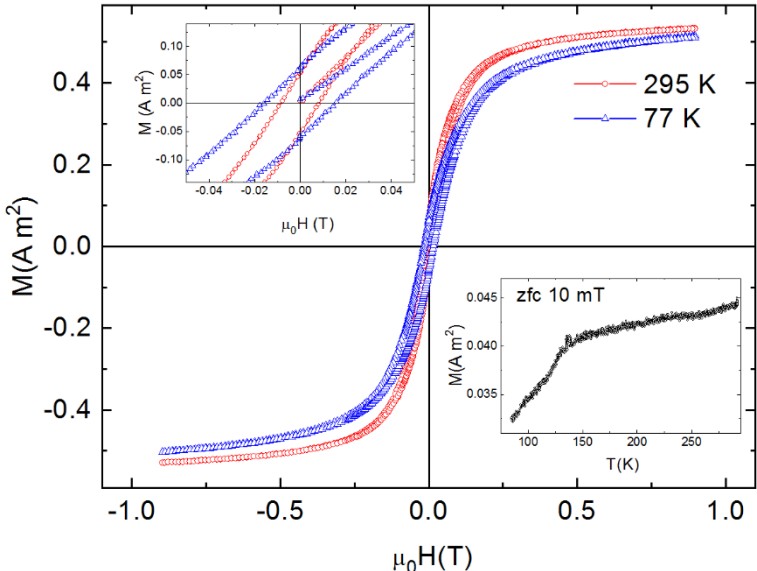

**Fig. 7. Magnetic hysteresis loops up to 0.9T at room temperature (red) and at 77K (blue). Top-left inset: close-up of the area close to the zero field. Inset bottom right: ZFC curve between 300K and 77K.**

In our recent contribution (Wilczyńska-Michalik et al., 2020a), the results of both magnetic and Mössbauer spectroscopy studies of the soil samples from sites at different distances from industrial pollution sources were presented. The magnetometric data previously reported for the soils show high degree of similarity to currently discussed samples.. In the case of Mössbauer spectroscopy, the situation becomes more complicated as the soil samples certainly undergo diverse processes due to oxygen exposure, humidity and so forth. However, still we observe that the fingerprints of anthropogenic particles found in our present study are clearly recognised in soils from polluted areas, in sharp contrast to soil samples from sites far from industrial plants.

### 3.6. Possible environmental and health impact of Fe-rich particles

Although direct environmental and health impact wasn't studied it can be suggested that both high abundance of Fe-rich particles and form of their occurrence (high content of nanoparticles) indicate potential threat. Finely dispersed Fe and other transition metals rich particles from various sources (e.g. combustion, friction, industrial emission, crustal material, road dust) are considered to have negative impact on human health (Lodovici and Bigagli, 2011; Maher et al., 2016; Calderón-Garcidueñas et al., 2019; Gonet and Maher, 2019; Maher, 2019; Maher et al., 2020; Shahpoury et al., 2021; Hammond et al. 2022).

Fe and other transition metals rich particles (especially when finely dispersed) can be active in reactions occurring in the atmosphere, e.g. in the catalytic oxidation of $SO_2$ in the atmosphere and formation of sulphate aerosol (Alexander et al., 2009; Dupart et al., 2012) or formation of brown carbon particles (e.g. Al-Abadleh 2021).

High content of anthropogenic Fe-rich particles in the studied material related to fuels combustion or other high temperature processes can suggest relatively high bioavailablity of Fe (e.g. Ito et al., 2019; Ito et al., 2021a). Determination of the wide

span of size of Fe-rich particles resulting from our study is also important taking into account their different potential reactivity (Liu et al., 2022).

Fe-rich, airborne particles participate in the heating effect in the atmosphere (Moteki et al., 2017; Ito et al., 2018) and impact climate changes. It is important to emphasis the role of particles' size and in  heating (Ito et al., 2021b).

## 4 Conclusions

- Magnetic components are present as discrete particles, as particles attached to larger grains or in aggregates of various size and composition.

- Fe-rich magnetic particles differ in morphology (irregular or spherical) and size (from above 20 $\mu$m to nanoparticles well below 100 nm). Apart from Fe, other metals are also present in these particles.

- Spherical particles formed in high temperature processes are of anthropogenic origin. Most of the irregular particles
are probably also of anthropogenic origin, but natural sources could also be considered.

- The abundance of spherical particles is higher among smaller particles. Spherical Fe-rich particles below 200 nm in size (often containing Zn and other metals) are a characteristic component occurring often as homogenous aggregates.

- The results of the XRD studies suggest that separated fraction is dominated by magnetite, hematite and $\alpha$-Fe in terms of the magnetic phases. It also exhibits quartz, feldspar and pyroxene.

- Mössbauer spectroscopy indicates the presence of $Fe_3O_4$, $\alpha$-$Fe_2O_3$, metallic Fe (FeX alloy) and $(Ca,Mg,Fe)(SiO_3)_2$. Clear evidence of the occurrence of nanometre scale $Fe_3O_4$ particles was shown.

- The study of a larger number of samples will give the possibility of better understanding the range of variability of the material as well as the health and environmental impact.

- Results indicate that analysed Fe-particles can impact human health and environment in multiple ways in urbanized
areas.

- Studying samples obtained directly at the emission sources and their comparison to our present results could give indication of the impact of the particular industrial activities on the environment.

**Acknowledgments.**

The Ministry of Science and Higher Education by the Faculty of Physics and Applied Computer Science AGH UST statutory
tasks (Ł. G., J. M. M., W. T and J. Ż.), the Faculty of Geography and Biology at the Pedagogical University of Kraków (W.W.-M.), and the Faculty of Geography and Geology at Jagiellonian University (M.M.) funded this research. The study was included in "The Anthropocene as the Epoch of Natural Environment Transformation" project at Pedagogical University. At

Jagiellonian University, the study was performed within the "Anthropocene" Priority Research Area under the "Excellence Initiative – Research University" programme. J.M.M acknowledges partial support by the National Science Centre  in the framework of the MINIATURA 5 founding (grant no.: UMO-2021/05/X/ST10/00975).

Authors are grateful to Waldemar Obcowski for his help in preparation of some figures.

Map in Figure 1 is modified on the basis of OpenStreetMap® open data, licensed under the Open Data Commons Open Database License (ODbL) by the OpenStreetMap Foundation (OSMF).

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
