# Peer review of "Magnetic fraction of the atmospheric dust in Kraków – physicochemical characteristics and possible environmental impact"

_EGUsphere, 2022_

## Author Comment (AC2)

SUPPLEMENTARY INFORMATION 1 – Sampling site, static magnetic sampler

[Figure]

*Figure SI-1-1. Sampling site (50.026916979306854 N, 19.902035577195356 E)*

According to Google Maps distances (straight lines) to places related to personal/public transport are as follows:

1. 125 m to the nearest tram lane (separated by at least 5 m tall noise barrier and 3 storey buildings with 35 m wide gap between them);

2. 142 m to the nearest two lane street (separated by at least 5 m tall noise barrier and 3 storey buildings with 35 m wide gap between them);

3. 35 m to the car park (12  parking spots approximately)

4. 40 m to the car park (30 parking spots approximately; mostly hidden on the other side of the 1,5 storey building)

5. 60 m to the car park (20 parking spots approximately)

6. 25 to the closest car park (20 parking spots @ 25 m distance, 20 parking spots @ 35 m, 20 parking spots @ 45 m  approximately)

7. 50 m to the car park (20 car parking spots and 8 bus parking spots approximately)

8. 50m to the car park (30 parking spots approximately)

[Figure]

*Figure SI-1-2. A: View of the sampler located at about 150-170 centimetres above the ground level in the grassy area at the III Campus of the Jagiellonian University (Gronostajowa Street, Kraków, Poland – exact location in the Figure SI-1-1); B: Closer view (area marked with yellow rectangle) of the magnet matrix under the cover placed in order to protect from precipitations.*

[Figure]

*Figure. SI-1-3. A: Simulation of magnetic field induction vector direction along a matrix of solid magnets; B & C: measured magnetic field intensity along the matrix of magnets (B) and in a direction perpendicular to the long axis of magnets – x, y and z directions of magnetic field as indicated.*

To collect the magnetic fraction of atmospheric dust, a static (passive) sampler composed of a matrix of solid magnets arranged to increase gradients and magnetic field strength was used (Fig. SI-1-3 A, B, C). It was covered with a 25 μm thick PVC film in order to ease the separation of the collected sample from the sampler itself.

[Figure]

*Figure SI-1-4. Magnetic fraction on the passive sampler after 9 months of deposition; A, B & C. Sample on the surface of rectangular magnet covered with PVC foil with dust grains of various colour, size and shape on the surface of the sampler. C&B are zoomed areas marked by yellow rectangular areas in A&B respectively.*

---

## Author Response (AR1)

EGUsphere, author comment AC1
https://doi.org/10.5194/egusphere-2022-462-AC1, 2022
**Reply on RC1**

Jan Marek Michalik et al.
* * *
Author comment on "Magnetic fraction of the atmospheric dust in Kraków – physicochemical characteristics and possible environmental impact" by Jan Marek Michalik et al., EGUsphere, https://doi.org/10.5194/egusphere-2022-462-AC1, 2022
* * *
Authors are very grateful for his/her work on reviewing our submission and for his/her positive report. The changes suggested by the Reviewer as "technical comments" are being included in the revised version of the manuscript. Below we present our response to points raised by the Reviewer as "Specific comments".

*1) More details in the Method section would be valuable, e.g.:*

*1a) [Lines 58-63] More info re sampling site. Is it next to a busy street? How far from the street curb? How busy this street is? How many lanes? What is the traffic volume? Etc.*

According to Google Maps distances (straight lines) to places related to personal/public transport are as follows:
I. 125 m to the nearest tram lane (separated by at least 5 m tall noise barrier and 3 storey buildings with 35 m wide gap between them);
II. 142 m to the nearest two lane street (separated by at least 5 m tall noise barrier and 3 storey buildings with 35 m wide gap between them);
III. 35 m to the car park (12 parking spots approximately)
IV. 40 m to the car park (30 parking spots approximately; mostly hidden on the other side of the 1,5 storey building)
V. 60 m to the car park (20 parking spots approximately)
VI. 25 to the closest car park (20 parking spots @ 25 m distance, 20 parking spots @ 35 m, 20 parking spots @ 45 m approximately)
VII. 50 m to the car park (20 car parking spots and 8 bus parking spots approximately)
VIII. 50m to the car park (30 parking spots approximately)
We have decided to add a figure with a situation map (attached) and not a list like the one above, but we are opened for suggestions.

*1b) [Lines 70-80] It is unclear how the samples were collected and dispersed in isopropanol. Was the PVC foil ultrasonicated in isopropanol? For how long?*

Outer layer of the PVC foil was carefully cut through one of the edges. As the material was mostly stuck to its surface (importantly the vast majority of thin blackish phase was so) is was sonicated for 15 minutes in IPA, then carefully washed with IPA keeping the volume of the liquid as low as possible in order to control the sample in easier way.

*1c) [Lines 82-88] What kind of powdered samples? Were they collected separately from these collected with magnets and PVC foil?*

The sentences could have been misleading. First (82) refers to the fact that these were powder - ground in order to make them homogenous in terms of gran sizes samples. As this fact shall be obvious the sentence can be removed. Starting form (85) it refers to the fact that the samples have been milled in order to separate Fe (magnetic) containing phase form surrounding matrices.

*2) Some discussion on advantages and disadvantages of this kind of sampler would be interesting. There are obvious benefits of such a system (e.g. simplicity, low workload required, etc.) but also some drawbacks (e.g. very poor temporal resolution, dependence on the weather, etc.). From a practical side, how big is the system? How much material (mass? volume?) was collected in the 9 months of sampling? This kind of discussion would be interesting regarding the potential use of such passive sampler instead of 'active' vacuum-based PM10/PM2.5 samplers.*

We agree with the following issues raised by the referee: the temporal resolution is very poor. In fact 9 months of collection can cover up some seasonal variations. However particles we are tracing are formed in high temperature industrial processes and winter/summer differences are not that straightforward as in the case of soot just to give an example. Then we designed the sampler in the way it is not affected by rain/snowfall (at least it "active" part).
Practical aspects
- Size: 50 x 100 mm (for the "collecting area", but that would depend on the single magnet size. Given the results I would say that keeping the magnets as small as possible in order to have better "single magnet edge" to "single magnet surface" ratio and thus better field gradients shall lead to better results. However with smaller magnets mechanical stability (bending for example) would be poor so the overall surface shall be smaller. Then the rest of the design can be adapted to the needs.
- The volume of the material is really small. In fact part of the material was used for chemical analysis and magnetic measurements so was either lost in process or partially lost in holders. It was also far to small quantity for easy sample preparation for Mossbauer spectroscopy (filler was needed to add volume). Also SEM/EDX analysis require the material to be used just for that purpose. For next study / as a suggestion I would propose somehow bigger active area.

*3) [Lines 179-198] Interesting discussion on the sources of the magnetic particles. It is indeed challenging to unambiguously determine the source of these particles (it is most probably a mixture of several sources). High concentration of Cu and Zn in some of the particles might suggest the non-exhaust vehicle sources (brake- and tyre-wear). Was Ba analysed? Its presence would also suggest the brake-wear as the source.*

We are currently working on the follow-up study related to the brake-wear particles, coal combustion etc. Certainly it is a challenging task but we believe that based on the data

presented in the manuscript under consideration, large scale ongoing and planned studies, recent results and available published data we can elaborate conclusions regarding the magnetic micro- and nanoparticles. Also I would like to comment that the manuscript under revision may serve as an invitation for collaboration for groups studying similar problems but having more specific experience (e.g. metallurgy, road transportation, rail transportation, aerosols in general, etc.)

The presence of Ba is mentioned in line 223. It is usually accompanied by S and it can suggest that it is related to coal combustion. The highest measured Ba content was 4.82wt%; usually is lower.

3a) [Lines 199-205] Well, high concentrations of Fe oxides can be also derived from vehicular brake systems, as shown in several recent studies so it would be also worth considering this potential source as the concentrations of magnetite (and other Fe-rich minerals) in brake-wear PM can be even an order of magnitude higher than in emissions from industrial processes! Also, more details on the sampling site location would provide more information on this topic (cf. comment 1 on the Method section).

The possibility of the emission of Fe-rich particles from brake-wear is mentioned in line 197. Without detail results of chemical and morphological analysis of brake-wear origin particles we withheld the discussion.

We would also like to add tha table with EDX analyses results (.pdf file attached) as a Supplementary Information.

Please also note the supplement to this comment:
https://egusphere.copernicus.org/preprints/2022/egusphere-2022-462/egusphere-2022-462-AC1-supplement.pdf

[Figure]

EGUsphere, author comment AC4
https://doi.org/10.5194/egusphere-2022-462-AC4, 2022
**Reply on RC3**

Jan Marek Michalik et al.
* * *
Author comment on "Magnetic fraction of the atmospheric dust in Kraków – physicochemical characteristics and possible environmental impact" by Jan Marek Michalik et al., EGUsphere, https://doi.org/10.5194/egusphere-2022-462-AC4, 2022
* * *
In my opinion, the work is poorly organized, which makes it difficult to read. First of all, the authors did not include the purpose of the work, which makes its evaluation much more difficult. It is not clear what the authors wanted to focus on and, as a result, what they wanted to show.
*The aim of the study was to characterize magnetic fraction of aerosols in Kraków. Collection of the analytical material is very important in such study and the simple passive sampler was prepared and presented in the manuscript. Suitable explanation related to the aim of the study will be added in the text.*

Introduction section - there is no presentation of research results on the magnetic fraction of various types of dust.
*Data related to different types of magnetic dust are presented briefly in the discussion for comparison with our results. In our opinion "general review" of magnetic fractions of various types of dusts in the Introduction would result in repetition of data and significant extension of the text.*

Methods section - it is required to complete the information on the research area (reasons why Krakow was chosen), detailed information on the method of collecting samples, and some details on the measurement methods.
*This subject was already raised by the referees. In "Reply on RC1" we have agreed, that more information is needed. Consequently a map will be added to the final version of the manuscript. In "Reply on RC2" we have included the "Supplementary Information 1" ("SI_1") where the sampling area is described as well as some more information on the sampler. This will be extended with the explanation given in "Reply on RC1". We find it is better not to include too much in the main manuscript as it moves the readers focus to the technical issues.*

In my opinion, the Result and Discussion section should be separated, the first section should present the results (data should not be included in the captions of the figures see Fig. 3 and 4).
*This was already said by other referees and we include all the mentioned results in the "Supplementary Information 2" ("SI_2") – see "Reply on RC1".*

Discussion section - here the results of the research should be discussed in accordance with the purpose of the work.
I agree with most of the comments of two earlier Reviewers, so I will not repeat them.

Other commands:
It should be remembered that the magnetic fraction of dust, in fact, are mainly ferromagnetic minerals, sensu lato, with relatively strong magnetic properties. The remaining particles (after separation) also have magnetic properties: para- or dia-. So, in this method of separating magnetic particles, do we get rid of particles that exhibit paramagnetic properties? This issue was not discussed.
*We will add the discussion in the final version of the manuscript.*

l. 28 how to understand "dust fall samples"
*The term "dust fall samples" is widely used and defined in 60-ties (DOI: 10.1080/00022470.1966.10468490), it is commonly used nowadays (e.g. cited papers by Liu et al. 2019; Magiera et al. 2010) to describe deposition of dust from the atmosphere.*

l. 35 what size is considered to be magnetite nanoparticles
*The referred line corresponds to the cited reference (Zhang et al. (2020), Zhang Q., Lu D., Wang D., Yang X., Zuo P., Yang H., Fu Q., Liu Q., Jiang G., 2020. Separation and Tracing of Anthropogenic Magnetite Nanoparticles in the Urban Atmosphere. Environ. Sci. Technol. 54, 9274–9284. https://doi.org/10.1021/acs.est.0c01841. In that paper we find what follows:*
*„Interestingly, we find that the magnetite NPs extracted from PM2.5 highly resemble the mixture of road traffic- and CFAderived particles (see Figure 3a,i,n). Particle size distribution obtained by nanoparticle tracking analysis (NTA) shows that the road traffic-derived magnetite displays a relatively broad distribution from 80 to 230 nm with two feature peaks at 126 and 177 nm (Figure 3s), probably resulting from the two distinct origins of the particles (i.e., combustion and noncombustion sources). The CFA-derived magnetite exhibits a narrow size distribution from 80 to 120 nm with only one feature peak at 97 nm (Figure 3t), which is consistent with the result obtained in TEM (Figure 3n). Noteably, the PM2.5- derived magnetite shows three feature peaks at 100, 122, and 200 nm, which may correspond to the sum of the particles derived from road traffic and CFA (Figure 3u). We infer that the peak at 200 nm may shift from the peak at 177 nm probably due to some minor sources or particle aggregation during atmospheric transfer process.*
*It is worth noting that NTA is a light scattering method and evaluates the particle diameter on the basis of tracing the Brownian motion of particle diffusion.42 The best working range of NTA for particle size analysis is 50−1000 nm, because particles smaller than □50 nm scatter much less light, so smaller particles tend not to be detected; meanwhile, for particles larger than □1 μm, their Brownian motion is too small to allow their diameters to be extracted from the images. Thus, although a number of small particles (<50 nm) are observed in TEM (Figure 3a), they are omitted in NTA. Furthermore, the presence of aggregates of small magnetite particles may also be counted as large particles (Figure S11). Therefore, for comparison, we have also performed the particle size statistics based on TEM counting. The results are given and discussed in Figure S12. Comparing the results between TEM and NTA, TEM gives more information in the small size range, while NTA, as a bulk measurement technique, can cover all particles in a sample to avoid possible bias caused by particle selection. In addition, the particle size obtained by NTA is hydrodynamic in diameter, which is normally larger than that obtained by TEM.43 In spite of the discrepancy mentioned above, the profiles of particle size distribution obtained by these two techniques are largely similar."*

l. 43 PM is an abbreviation; particulate matter
*"filters containing PM10 particulate matter" was used in the text (line 43) to indicate that PM10 fraction of the total particulate matter occurring in the atmosphere was analysed. For simplicity we will cancel "particulate matter" in the final text.*

l. 262-263 In Magiera et al. 2021, I did not find information "possibility of superparamagnetic particles being present in the suspended particulate matter was

discarded by some authors on the basis of the magnetization measurements"

*In cited reference the authors say (page 9, second but last paragraph):*
*"There is also a theoretical possibility that some contribution to this component is made by fine magnetite particles that are close to the SD/superparamagnetic boundary, which has clearly reduced hyperfine field due to spin relaxation effects. However, this result was in contrary to the magnetic measurements (low χfd%). In general, the precise distinguishing among the phases ascribed to the G06 component is impossible due the broad overlapping lines and the similarity of the hyperfine parameters."*

*Then, in the same reference (page 12, first full paragraph):*
*"Very low values of frequency dependence of magnetic susceptibility, usually less than 2.0%, suggests that the content of superparamagnetic grains is negligible. Only areas influenced by long-range transport from other industrial sources (power plants coke plant) located in long-distance, exhibited the considerable admixture of SD particles and higher χfd% values (>4.5)."*

*Consequently we propose to change the sentence in lines 266-270 from:*
*"On one hand, a possibility of superparamagnetic particles being present in the suspended particulate matter was discarded by some authors on the basis of the magnetisation measurements (Magiera et al., 2021). On the other, the possibility of the long-range transportation of tiny (a few nanometres to a few tenths of a nanometre in diameter) particles far from the industrial sources cannot be eliminated."*
*To:*
*"On one hand, a negligible content of superparamagnetic particles in the suspended particulate matter was reported by some authors (e.g. Magiera et al., 2021) on the basis of frequency dependence of magnetic susceptibility analysis. On the other, the possibility of the long-range transportation of tiny (a few nanometres to a few tenths of a nanometre in diameter) particles far from the industrial sources cannot be eliminated, as was also proposed by the same authors."*

*What about interpretation of curves zero field cooled (ZFC) and field cooled (FC)?*
*Typically, a well-defined maximum in the ZFC curve is associated to blocking temperature of superparamagnetic nanoparticles of well-defined size. However, for broad distribution of nanoparticles' sizes such maximum is usually not observed. This is apparently our case, as the ZFC curve shows no maximum but constant raise up to the room temperature. Appropiate discussion will be added.*

Several references cited in the text are not included in the references section.
*This issue will be addressed in the final version of the manuscript.*

[Figure]

EGUsphere, referee comment RC4
https://doi.org/10.5194/egusphere-2022-462-RC4, 2022
**Reply on AC2**

Anonymous Referee #2
* * *
Referee comment on "Magnetic fraction of the atmospheric dust in Kraków – physicochemical characteristics and possible environmental impact" by Jan Marek Michalik et al., EGUsphere, https://doi.org/10.5194/egusphere-2022-462-RC4, 2022
* * *
*We are grateful for the Referee's effort in reading our submission and his/her suggestions on improving the quality of the manuscript. We are addressing the questions/remarks/suggestions below. The changes will be included in the revised version of the manuscript and/or separate Supplementary Information document when we find it more adequate.*

- It is unclear what the purpose of the research presented was: whether the purpose of the manuscript is to present a novel way to sample the magnetic fraction or whether the purpose of the manuscript is to thoroughly characterize the magnetic fraction of dust found in the urban atmosphere. In the first case, the manuscript needs to be supplemented at least with a detailed description of the device's design and, finally, with an evaluation of its performance based on a comparison of the results with those from other methods as well as a recommendation for further research. Otherwise, it should be justified that this single sampling site is representative of the content and composition of magnetic fraction in the air.

The purpose of the work should be defined in the last paragraph of Introduction section, which is missing here.

*The aim of the study was to characterize magnetic fraction of aerosols in Kraków. Collection of the analytical material is very important in such study and the simple passive sampler was prepared and presented in the manuscript.*

*Single sample collected during nine months period gave a time averaged information. Sampling site was situated far from important sources of emission (e.g. industrial plant) and the sample can be considered as a "background" for the urban area. There are no indices of significant spatial variation in the aerosols composition (excluding close vicinity of individual sources of emission).*

*Testing the performance of the sampler was not the aim here, however we assume (which is supported by the discussions during seminar presentations of the results) that the way of sample collection we propose is interesting. On one hand it as cost effective (as compared to aspiration with cyclonic filters) and on the other all collected material is available for research (which is not the case when fibrous filters or membranes are used).*

*Suitable explanation related to the aim of the study will be added in the Supplementray*

*information (see attached file).*

**Thank you for explanation. Please include this explanation in the manuscript. Do not include the aim of the study in Supplementary information. Please amend the Introduction with properly edited last paragraph and include the information you have provided above. You can follow the good examples below which are taken randomly from the literature cited in your manuscript.**

Baker, A.R., French, M., Linge, K.L., 2006. Trends in aerosol nutrient solubility along a west–east transect of the Saharan dust plume. Geophys. Res. Lett. 33. https://doi.org/10.1029/2005GL024764

In this study we examine longitudinal variations in the solubility of aerosol Fe, P and Si using samples collected during Meteor cruise 55 (M55), a west – east transect of the tropical Atlantic Ocean in October/November 2002 [*Wallace and Bange*, 2004], and we include aerosol Al and Mn solubility data for comparison with our results from other regions of the Atlantic [*Baker et al.*, 2006]. We use simple leaching protocols to investigate whether atmospheric processing has a significant effect on aerosol nutrient solubility over the spatial/temporal scales associated with transport of Saharan dust across the Atlantic Ocean. We emphasise however that these leaching protocols do not directly reproduce the solubility behaviour of aerosols on deposition to seawater, as discussed by *Baker et al.* [2006]. The M55 cruise track also crossed the Intertropical Convergence Zone (ITCZ) on a number of occasions and this enables us to compare nutrient solubility in Saharan dust aerosol with aerosols which originated in the southern hemisphere.

Rutkowski, R., Bihałowicz, J.S., Rachwał, M., Rogula-Kozłowska, W., Rybak, J., 2020. Magnetic Susceptibility of Spider Webs and Dust: Preliminary Study in Wrocław, Poland. Minerals 10. https://doi.org/10.3390/min10111018

The application of spider silk in the magnetic monitoring of urban road dust has never been directly compared to conventional methods. This study aims to report the magnetic susceptibility of urban road dust settled in indoor and outdoor environments, and to compare the data with results obtained from spider webs exposed to indoor and outdoor pollutants. The research hypothesis states that spider webs are appropriate substrates for monitoring environmental pollution due to their good entrapment of aeolian particulate pollution, and thus, they can be employed in studies of urban pollution pathways and human exposure to pollutants. Additionally, road dust represents an equilibrium of deposition and erosion since the last rain, wind, or storm event. Thus, we can assume that spider webs can re-disperse aeolian dusts in a linear and proportionate way, similar to stratigraphic layers of sediment on a road, allowing relationships to be established between webs and road dust and between indoor and outdoor sediments.

Moteki, N., Adachi, K., Ohata, S., Yoshida, A., Harigaya, T., Koike, M., Kondo, Y., 2017. Anthropogenic iron oxide aerosols enhance atmospheric heating. Nat. Commun. 8. https://doi.org/10.1038/ncomms15329

In this study, *in situ* aircraft measurements using a modified single-particle soot photometer (SP2)[25] and electron microscopy are performed to show that anthropogenic $FeO_x$ particles, particularly aggregated magnetite nanoparticles, are ubiquitous in the continental outflows from East Asia. We then evaluate their contribution to atmospheric shortwave absorption on the basis of the observed size-resolved mass concentrations and

particle morphologies. Our results indicate that the absorption by anthropogenic FeO$_x$ is at least 4–7% of the BC absorption over East Asia.

- Introduction

  The Introduction lacks a final paragraph stating the goals/objectives of the work. Also, the Introduction lacks a paragraph discussing potential sources of magnetic particles (which is discussed in the Results section, among others).

  The paragraph related to the main goals will be added to the text.

  I suggest rewriting the Introduction section providing the information in the following order: characteristics of magnetic fraction in air dust, known sources, known health effects, known environmental effect, various methods of sampling, and then the "last paragraph of Introduction".

*The Introduction contains information related to approaches used in magnetic fraction of aerosols studies (lines 23-38), health (lines 39-43) and environmental effects (lines 49-55). Sources are not discussed in the Introduction. Discussion is included in the chapter Results and discussion to avoid repetitions of data and citations.*

**No comments.**

C. Conclusions

Please comment on the novelty of the sampling method proposed. Would you suggest wide application of this design and methodology? What are pros and cons?

*Sampling method used in the study is relatively simple and inexpensive. It is possible to collect a sample significantly enriched in magnetic components. Relatively high content of non-magnetic particles is most probably related to the presence of complex aggregates of various particles in the atmosphere. Simplicity of the sampling method can be considered as suggestion for its wide application. On the other hand it would be interesting to test other samplers of magnetic fraction. Also, it is worth mentioning that in contrast to other commonly used "active" methods is that only the particles really suspended in the air are stuck on the magnet surface. We are of course aware that strong wind conditions can affect the efficiency of the sampler and change the composition of the samples. However long collection time, and elimination of rain induced deposition are certainly an advantage.*

**Please include this explanation in the manuscript in Conclusion section.**

- The characterization and discussion of the results is strongly biased toward spherical particles. Please acknowledge the presence of angular particles derived simply from rust. The word "rust" is not even used in the text. Rust is probably the main source of Fe-rich particles in urban environment. What is their relation to the magnetic fraction? Are you sure no hematite or goethite particles accompany the magnetic fraction? Please supplement the text with relevant observations, comments, and literature data.

*Spherical particles are really an important point of the discussion of electron microscope study. It is related to the anthropogenic, high temperature origin.*

*Rust is commonly occurring (on rails, cars and other vehicles construction elements, urban infrastructure, etc.). It can be considered as a source of Fe-rich particles (e.g. hematite particles). This interpretation will be added to the text. Goethite has not been identified in our sample.*

**Please include this explanation in the manuscript.**

Specific comments:

Line 49 – chemical formula of SO2 needs correction

Line 57: Magnetic fraction collection

- please expand and rephrase the description by stating, among other things, what size the individual magnets were, how many of them, what the total collecting area was, why the sampler was placed in such a way (vertically, at a height of 1.5-1. 7m), why the sampler was placed in such a location (whether the location was chosen deliberately because of existing knowledge of air pollution in the city or because of convenience), why collection lasted 9 month, which part of the year (seasons), whether the surface was protected from precipitation from above, whether the direction of the vertical collecting surface was directed in the direction of the most common wind direction, how far the grass-covered ground surface reached, whether there are potential sources of magnetic particles in the area, how far away were potential close-range sources (e.g. streets, streetcar lines) and long-range sources, etc. Is it possible to show a photo of the sampler or installation?

*Data will be added in Supplementary Materials (see SI_1).*

- The information provided in Figure 1 is not discussed or explained in the text. Why were magnetic field measurements and simulations performed, how do they compare, what is their relevance to the predicted performance of the sampler design, how do these measurements and simulations support the predicted performance of the sampler, to what extent was this confirmed by the observed distribution of magnetic particles on the PVC film (Fig. 1 D and E), etc.?

**Please answer these questions. Please include the answers in the manuscript. Please use the information provided in Figure 1 in the text.**

Line 83

The powdered samples were placed on single-crystalline silicon no-background holders.

Does this mean that the sample was ground before analysis, or was it analysed as is?

The sample was ground prior to the analysis.

**Please include this explanation in the manuscript.**

Lines 85 – 87

This separation process, for obtaining a laboratory concentrate of the magnetic fraction, is no different from obtaining the magnetic fraction from dust samples collected with classical samplers. Please include such a comment in the text, here or in Conclusions, so that it is clear that for this analytical method the proposed way of sampling atmospheric dust did not give positive effect.

*Separation of magnetic components from significantly enriched in magnetic particles sample is very effective. It is possible to consider two "classical" methods of aerosols collection. Separation of magnetic fraction from filters (e.g. quartz filter) in sampling using aspirators is difficult. Samples obtained using this method are very rich in combustion products (soot, tar balls). Samples collected by sedimentation (dust fall samples) contain material from dry and wet deposition. Samples of dust fall contain often big quartz grains. Magnetic fraction is a very minor component in dust fall by weight and its separation is much more difficult.*

**Please include this explanation in the manuscript.**

Line 87

The collected diffraction patterns were analysed in terms of the Rietveld method using the FullProf Suite Package (Rodríguez-Carvajal, 1993).

What does it mean: phase identification or quantitative analysis, or both? Please add in the text.

*FullProf suite was solely used for Rietveld refinement, which resulted in quantification of observed crystalline phases. The phase identification was a multistep process, which involved: analyses of chemical composition followed by trial refinements of the most apparent, common phases; usage of automatic phase recognition software (eg. Panalytical HighScore 3.0) for search of secondary phases and confirming the main phases.*

**Please include this explanation in the Methods section.**

What was the resolution of the collected diffraction pattern? Was it sufficient for proper analysis by the Rietveld method? Please add in the text.

*The resolution of the instrument calibrated using NIST 660 standard is 0.065° of 2θ, which is much below widths of lines in samples. The data were collected with angular step of 0.016° of 2θ. The calibration measurements using NIST 660 revealed also instrumental broadening of the collected lines, which is essential to further analyses of reflections profiles.*

**Please include this explanation in the manuscript.**

Lines 90 – 96

Please provide the following information in the text:

Did you use a separate subsample of the material collected on the PVC film, different from the sample used for XRD?

*We first used the sample for Mossbauer spectroscopy. Then the filler (sucrose) was removed and the sample was measured with x-ray diffraction. As the SiO2 peaks dominated over other reflections we opted for separating the magnetic species once more. This allowed removal of silicon oxide matrix embedding iron compounds that were of the main interest.*

How did you split the sample to ensure its representativeness?

Did you use a method of concentrating the magnetic fraction similar to that used in XRD measurements? What was the size/mass of the sample used for magnetization measurements? Was it sufficient to make the measurement? Was it a separate subsample, or material previously used for other measurements? Please provide this information in the text.

*All of the collected material (apart from a small amount used for SEM/EDX) was used for magnetization measurements. We could have used some subsample but assuring representativeness would be difficult and the measurements could have taken much longer time.*

**Please include all these explanations in the manuscript to clarify the methods. Apparently, this is important information in case someone tries to duplicate this approach, since the order of analysis is crucial to success.**

Line 99: What was the total mass of material collected over a period of 9 months on the ... mm2 of the sampler?

*The sample was not weighted. In fact the amount was so small the Mossbauer spectroscopy analysis seemed impossible without adding a filler. Even though the measurements lasted longer than usual until reasonable statistics were achieved especially at liquid nitrogen temperature.*

**Please include this explanation in the manuscript to explain why it was not possible to estimate the amount of magnetic dust fallout.**

Line 104

Instead for:

The results of the XRD studies (Fig. 2) suggest that the separated fraction

There should be:

The XRD results (Fig. 2) suggest that the magnetic fraction separated from the collected sample

Lines 106 - 107

Precise analysis of the profile of magnetite reflexions in the XRD pattern suggest the distribution of various elements at the Fe-sites (e.g. Cr, Mn, Co, Zn) as typical of naturally abundant ferrites (Fig. 2).

Please elaborate, it is not clear from this sentence which part of the curve and which observation or which part of Fig. 2 leads to this conclusion.

*This remark was based on evidenced strain in the profile of this phase. The strain could be extracted from instrumental broadening (thanks to calibration measurements), therefore Rietveld refinement revealed strains for all phases. The residual strains are related to static defects of the structure e.g. atomic disorder on Wyckoff positions originating from substitution of different types of atoms into specific sites. The refined strain was found to be about 0.75%, which is significantly higher than typical values for pure Fe3O4 specimens which is smaller than 0.2% (The pure Fe3O4 specimen was also measured as a reference). The resulting broadening of the main reflections of the Fe3O4 (220 at 30.1° of 2θ and 311 at 35.5° of 2θ) can be noticed in the inserts to the Fig. 2.*

**Please include all these explanations in the manuscript.**

Lines 112 – 114

Please move this part of the text (discussion of SEM results, grain size and morphology) to line 103, so that SEM results are together, followed by XRD results.

The paragraph was organized according to material discussed (not according to method applied). Form of occurrence of non-magnetic components was discussed after information about their identification using XRD. The paragraph will be re-organized.

Line 215

Instead for Fe- rich particle It should be Fe- rich particles

Line 236

Domains with the magnetite ordering reach the size of 10 nm (Fig. 5G, H).

Please elaborate and explain what information relevant to the topic of this study was gained from HRTEM analysis and results presented in Fig. 5H? magnetite was identified already using several other methods.

*HRTEM imaging allows for proving the existence of the ferrous nanoparticles: 10 nm and smaller on one hand and bigger - up to 200 nm on the other. The former are small enough to cross human body barriers and the latter ones are more difficult to be removed from the human body. Moreover we show crystallinity of those smallest particles proving their chemical composition what is important form the point of view of the assessment of the health impact.*

**Please edit and then include all these comments in the manuscript (here or in the Methods section) to explain what information relevant to the topic of this study was gained from HRTEM analysis and results presented in Fig. 5H.**

Please move Figure 5 to a place below the text discussing its content.